# A Fast and Efficient Measurement System for Nuclear Spin Relaxation Times in Atomic Vapors

**DOI:** 10.3390/s19224863

**Published:** 2019-11-08

**Authors:** Ting Huang, Cunxiao Miao, Shuangai Wan, Xiaoqian Tian, Rui Li

**Affiliations:** 1School of Mechanical Engineering, University of Science and Technology Beijing, Beijing 100083, China; 15607937886@163.com (T.H.); lirui@ustb.edu.cn (R.L.); 2Beijing Automation Control Equipment Institute, Beijing 100074, China; wsajishe@163.com (S.W.); xiaoqiantian003@gmail.com (X.T.)

**Keywords:** relaxation times, nuclear magnetic resonance (NMR), optical detection, optical pumping, auto-test system

## Abstract

With the rapid progress of cutting-edge research such as quantum measurement technology, nuclear magnetic resonance (NMR) gyroscopes represent a major development direction of high-precision micro-miniature gyroscopes, which have significant advantages such as high precision, small size, and low power consumption. It is meaningful to measure the relaxation times of noble-gas atoms which are crucial indicators to accurately and quickly characterize the vapor cell performance as a core component of gyroscopes. In this paper, a test platform for relaxation time is built and an automatic relaxation time test system based on free induction decay (FID) and the π pulse method is designed to accelerate the relaxation time test. Firstly, the formula of the atomic dynamic process based on the Bloch equation was deduced, a GUI (Graphical User Interface) simulation based on the derived differential equation was conducted, and the moving process of the magnetic moment was visually described. Then, the virtual instrument was used to integrate multiple test instruments into an auto-test system, and LabVIEW programming was used for control to realize the automation of the test process on the test platform. Finally, the test results in different conditions were compared. The results show that the test system is stable and reliable with excellent man–machine interaction, and the measurement efficiency was increased by about 185%, providing an effective test scheme for vapor cell performance.

## 1. Introduction

Inertial navigation systems are widely used in aerospace and other fields, with the advantages of not being disturbed by the outside world, strong autonomy, and good concealment [1]. As the core component of inertial navigation, the precision of a gyroscope directly affects the precision of navigation. Atomic gyroscopes developed based on quantum sensing technology are a research hotspot in research institutions around the world due to their ultra-high theoretical accuracy [2,3]. This discussion focuses on a specific type of atomic gyroscope, the nuclear magnetic resonance (NMR) gyroscope, using the invariance of nuclear spin resonance frequency in the inertial space to detect shell rotation speed, with the significant advantages of high precision, small volume, and low power consumption [4,5].

In NMR technology, the nuclei of atomic complexes recover to their equilibrium state in a non-radiative way after the removal of the radiofrequency field, and the relaxation times are important characteristic constants of the relaxation process. The inert gas atoms contained in the vapor cell of the NMR gyroscope are used to measure angular velocity [6]. The relaxation times of inert gas atoms are important characteristic measurements of the performance of the NMR gyroscope, which are closely related to the signal-to-noise ratio (SNR) and angular random walk (ARW) [7], and they determine the detection sensitivity of the gyroscope. In the case of short relaxation times, the expansion rate of the precession of inert gas atoms is faster, the uncertainty of net spin is greater, and the measurement accuracy is lower. ARW accumulates from previous measurement results to existing measurement results, that is to say, the total error of a single measurement depends on the cumulative error of previous measurement values, and even a slight decrease in accuracy will significantly increase ARW. Therefore, it is important to measure the nuclear spin relaxation time accurately and rapidly for measuring the performance of the gyroscope and improving the detection sensitivity [8].

Section 2 introduces the testing principle in detail and simulates the dynamic process according to the principle. Section 3 briefly introduces the hardware of the relaxation test system and the automatic test software based on LabVIEW. Section 4 evaluates the running example and test results of the proposed system. Finally, we summarize the paper in Section 5.

## 2. Vapor Cell Testing Theory and Visual Simulation

### 2.1. Relaxation Time Testing Theory

An external constant magnetic field $B_{0}$ is applied along the Z-axis, and then Zeeman level splits along the magnetic field direction. According to the principle of statistical physics, the number of atoms in the Zeeman level follows the Boltzmann distribution, and the number of particles in each energy level is basically the same. Under the wavelength properties of the nuclear material, circularly polarized light spreads along the $B_{0}$ direction, which determines the sensitive axis of the gyroscope. Circularly polarized light interacts with electrons in the atomic orbitals of alkali metals to transfer electrons to an excited state [9], which is unstable and causes spontaneous decay. Angular momentum is conserved under certain conditions, and electron angular momentum is transferred to the nucleus of noble gases to generate a relatively high level of net nuclear magnetization. By absorbing and emitting photons continuously, the atomic ensemble is finally in a single spin state, thereby realizing the polarization of the group and forming the macroscopic magnetic moment $M_{0}$ [10,11].

After light pumping, any possible signal generated by the precession of one atom is offset by the precession signal from the other atom in the opposite phase. Without an atomic group with coherent phases, the total gas sample cannot produce a measurable signal. Thus, we apply alternating excitation magnetic fields, $2B_{1}\sin\left(\gamma B_{0}t\right)$, along the X-axis, which can be decomposed into a left-hand circular polarization field and a right-hand circular polarization field [12]. The circular polarization field with the same precession direction as Larmor precession is selected to establish the rotating coordinate system $X_{1}Y_{1}Z$, and then the alternating excitation magnetic field becomes a constant static magnetic field in the rotating coordinate system. The precession magnetic moment originally in a random phase tends to be in the same phase, that is to say, the precession phase of the atomic ensemble is coherent, forming a uniform precession that can be measured. From the rotating coordinate system, it can be observed that the macroscopic magnetic moment turns from the Z-axis to the $Y_{1}$-axis at the angular frequency $\gamma B_{1}$ around the $X_{1}$-axis. According to the Bloch equation [13,14], the dynamic behavior of macroscopic magnetic moment in the magnetic field can be expressed as follows:(1)$$\frac{d\vec{M}}{dt}=\gamma \vec{M}\times \vec{B}-R\cdot \left(\vec{M}-\vec{M_{0}}\right),$$
where γ is the gyromagnetic ratio, $\vec{M}$ is the instantaneous magnetic moment, $\vec{B}$ is the total magnetic field, and R is the relaxation matrix.

The total magnetic field consists of the main field $\vec{B_{0}}$ and alternating excitation field $\vec{B_{1}}$, expressed as follows:(2)$$\vec{B}=B_{0}\cdot \vec{e_{z}}+B_{1}\cos\left(\gamma B_{0}t\right)\cdot \vec{e_{x}}-B_{1}\sin\left(\gamma B_{0}t\right)\cdot \vec{e_{y}},$$

The relaxation matrix can be expressed as follows:(3)$$R=\left[\begin{array}{ccc} \frac{1}{T_{2}} & 0 & 0\\ 0 & \frac{1}{T_{2}} & 0\\ 0 & 0 & \frac{1}{T_{1}} \end{array}\right],$$
where $T_{2}$ is the transverse relaxed time, and $T_{1}$ is the longitudinal relaxed time.

The transverse relaxation time $T_{2}$ represents the time when the transversal component $M_{xy}$ of the magnetic moment M decays to zero, whose main influencing factor is the magnetic field inhomogeneity, which is the residual magnetic field gradient in the atomic cell. The longitudinal relaxation time $T_{1}$ represents the time when the longitudinal component $M_{z}$ recovers to the equilibrium state $M_{0}$ from the non-equilibrium state, whose main influencing factor is the collision of inert gas atoms with the cell wall and other atoms.

Substitute Equations (2) and (3) into Equation (1), we can obtain the following differential equations:(4)$$\{\begin{array}{c} \frac{dM_{x}}{dt}=\gamma (M_{y}B_{0}+M_{z}B_{1}\sin\left(\gamma B_{0}t\right))-\frac{M_{x}}{T_{2}}\\ \frac{dM_{y}}{dt}=\gamma (-M_{x}B_{0}+M_{z}B_{1}\cos\left(\gamma B_{0}t\right))-\frac{M_{y}}{T_{2}}\\ \frac{dM_{z}}{dt}=\gamma \left(-M_{x}B_{1}\sin\left(\gamma B_{0}t\right)-M_{y}B_{1}\cos\left(\gamma B_{0}t\right)\right)-\frac{M_{z}-M_{0}}{T_{1}} \end{array},$$

It is assumed that, when the macroscopic magnetic moment is in the direction of $+Y_{1}$ axis, the alternating excitation magnetic field is removed, and the macroscopic magnetic moment in the XYZ coordinate system returns to the direction of *z*-axis along the helix. The signal detected by the detection laser is the free induction decay (FID) signal, whose decay rate depends on $T_{2}$, which can be calculated by measuring the time constant of signal attenuation [15,16]. The rotation of the macroscopic magnetic moment is determined by the switch of the rotation field, that is, the alternating excitation magnetic field, which is called a pulse. Since the rotation angle of the macroscopic magnetic moment is 90°, this pulse is also called the π/2 pulse [12,17]. According to the formula of rotation, we can obtain
(5)$$\gamma B_{1}t_{\pi /2}=\pi /2,$$

The length of time of the π/2 pulse can be expressed as follows:(6)$$t_{\pi /2}=\frac{\pi }{2\gamma B_{1}},$$

The length of time of the π pulse is twice as much as π/2 pulse, expressed as follows:(7)$$t_{\pi }=\frac{\pi }{\gamma B_{1}},$$

The gyromagnetic ratio of X 129e is $-7.4003\times 10^{7}\;{\mathrm{rad}\;T}^{-1}{\;s}^{-1}$, and the measured current is 8.4 mA. The coil constant of a three-dimensional magnetic coil is 24 nT/mA. According to Equation (6), we can get the estimate of $t_{\pi /2}=0.21\;s$, which is 0.11 s less than the value obtained by the experimental measurements. The current is kept constant so as to keep the magnetic field constant, and the time of excitation pulse is changed. The initial amplitude of the FID signal is measured to obtain a series of data points. The data points show sinusoidal distribution with the pulse time length, and the time point at the position of π/2 is obtained through fitting, namely, the time length of the $t_{\pi /2}$ pulse.

If a small alternating excitation magnetic field is applied along the X-axis, the signal whose growth rate depends on $T_{2}$ grows to a certain amplitude and becomes stable. After a period of time, the π pulse is used to reverse the xenon spin, which changes only the direction of the magnetic moment, leaving all other states unchanged. Over time, the xenon atoms collide with the alkali and return to a positive polarization, and they are then excited again by a small magnetic field. The half-life of xenon atoms $T_{1/2}$ is obtained by real-time monitoring of their transition from one polarization to the other, whose length depends on $T_{1}$, and the relationship between the two can be expressed as follows:(8)$$T_{1/2}=T_{1}\ln2,$$

This equation can be obtained through the following steps. Firstly, Equation (4) is simplified. We ignore the action of the excitation magnetic field, and this equation becomes
(9)$$\{\begin{array}{c} \frac{dM_{x}}{dt}=\gamma M_{y}B_{0}-\frac{M_{x}}{T_{2}}\\ \frac{dM_{y}}{dt}=-\gamma M_{x}B_{0}-\frac{M_{y}}{T_{2}}\\ \frac{dM_{z}}{dt}=-\frac{M_{z}-M_{0}}{T_{1}} \end{array},$$

We then assume the following initial conditions:(10)$$\{\begin{array}{c} M_{x}=0\\ M_{y}=M_{0}\sin\omega \\ M_{z}=M_{0}\cos\omega \end{array},$$
where ω is the initially included angle between the magnetic moment and the Z-axis.

The differential equation is then solved.
(11)$$\{\begin{array}{c} M_{x}=M_{0}e^{-\frac{t}{T_{2}}}\sin\left(\gamma B_{0}t\right)\sin\omega \\ M_{y}=M_{0}e^{-\frac{t}{T_{2}}}\cos\left(\gamma B_{0}t\right)\sin\omega \\ M_{z}=M_{0}\left(1+\left(\cos\omega -1\right)e^{-\frac{t}{T_{1}}}\right) \end{array},$$
when $M_{z}$ decays to zero, the relationship between the longitudinal relaxation time $T_{1}$ and time t can be expressed as follows:(12)$$T_{1}=t/\ln\left(1-\cos\omega \right),$$

Considering the role of persistent excitation and the relationship between initial angle ω and persistent excitation, we can get the time t, namely, the half-life $T_{1/2}$ relationship with $T_{1}$, which is shown in Equation (8).

### 2.2. Visual Simulation

To better illustrate the correctness of the above test theory, a GUI was used for a simulation according to Equation (3), and the simulation results are shown in Figure 1 and Figure 2. Figure 1 shows the magnetic moment motion progress in FID measurement. In the upper-left three-dimensional figure, the sparse curve outside the dense curve in the shape of a straw hat is the motion curve of the magnetic moment when we apply the π/2 pulse, and the dense curve is the free induction decay curve of the magnetic moment. The two-dimensional figure below shows the observed signal of magnetic moment motion along the *X*-axis, which has the same significance as the signal collected in our experiment. In the upper right, there are some parameters and trigger buttons used in our simulation experiment, including the main magnetic field, the continuous excitation magnetic field, pulse intensity, etc. The simulation was only done to visualize the magnetic moment motion, in which parameter setting is not meaningful.

Figure 2 shows the magnetic moment motion process in the π pulse measurement. The sparse curve outside the dense curve in the shape of the small dumbbell is the motion curve of the magnetic moment when we apply the π pulse, which reverses the magnetic moment. During the relaxation process of the magnetic moment along the Z-axis, we observe the motion of relaxation along the X-axis and find that the magnetic moment decays to zero and then gradually returns to a stable state along the Z-axis.

## 3. System Construction

### 3.1. Experimental Platform

The schematic diagram of the vapor cell relaxation test experimental platform is shown in Figure 3, which mainly includes an alkali vapor cell, a three-dimensional magnetic coil, a non-magnetic heater, a magnetic shield, a pump laser, a sense laser, and an NI computer.

The experiment adopted a 4-mm cubic borosilicate glass cell with no coating. The cell contained noble gas atoms X 129e and X 131e at 2 and 8 Torr, respectively, alkali atom Cs, buffer gas $N_{2}$ at 760 Torr, and so on. The external area of the cell included a magnetically free heater, which was used to heat the alkali atoms in the cell and promote spin hyperpolarization. A home-made independently developed non-magnetic heating unit reduced the disturbed magnetic field caused by electric heating, by means of the precise spatial symmetric arrangement of cables. The cell and heater were placed in the magnetic shielding barrel to shield the effect of the environmental magnetic field on the magnetic moment. The coefficient of attenuation by the magnetic shield barrel was 10^4^. We applied the reverse magnetic field gradient −15 nT/cm to compensate for the residual magnetic field gradient. The pumping light path was composed of a distributed feedback laser and polarizer. The working frequency of the laser was the resonance frequency of the Cs atom $D_{1}$ line (894 nm) and $D_{2}$ line (852 nm). The detection light path was composed of the distributed feedback laser, polarizer, analyzer, and detector. After the detection laser passed through the vapor cell, the analyzer divided the beam into two orthogonal polarization beams, which were amplified by the detector, and the procession signal was output. The pump laser and sense laser were American New Focus commercial lasers. The sense laser frequency was detuned from the optical resonance frequency, and the detuning value was 0.2 nm.

The whole experiment was carried out on an optical table, and the specific experimental equipment is shown in Figure 4.

### 3.2. Program Design Based on LabVIEW

According to the above principles and simulation, an automatic vapor cell relaxation time test system was developed based on LabVIEW/MATLAB hybrid programming. This section uses the flow chart in Figure 5 to briefly describe the system.

The realization of the system mainly included four steps: *z*-axis main magnetic field output, temperature acquisition and control, resonance frequency search, and data acquisition, storage, and processing.

The main magnetic field was controlled by the source measurement unit (SMU) PXIe-4145. All the boards used in the experiment came from National Instruments, which is headquartered in Austin, Texas, USA. We obtained the output function, configuration current, and other operations by calling the NI DC power function in LabVIEW, and we used the SMU to output a stable DC signal. We then produced the main magnetic field using a three-dimensional magnetic coil whose error was within 700 pT.

Temperature acquisition and control were realized by a digital multimeter PXI-4072 and waveform generator PXI-5441. The multimeter called the NIDMM function in LabVIEW to read the resistance value of the temperature sensor, and then converted it into the temperature value through the expression node; it then displayed the temperature in the waveform chart in real time. The waveform generator called the NI-FGEN function to output a sinusoidal voltage signal, which was amplified by the power amplifier and then lost to a heating plate to heat the vapor cell. The whole temperature control adopted PID control to realize closed-loop control.

The continuous sinusoidal voltage signal was output to the magnetic coil by a synchronous multi-function DAQ device with fixed frequency difference to generate a continuous excitation magnetic field. PXIe-4492, a data acquisition card, read the analog signal of the detector and converted the analog voltage signal detected by the photoelectric detector into a digital voltage signal. The program automatically intercepted the stable part of the signal, measured its amplitude, and compared and obtained the maximum amplitude value, whose corresponding frequency was the resonance frequency. The program output a specific pulse signal to the magnetic coil according to the resonance frequency found.

The analog signal output was read by the photoelectric detector, which was displayed in a waveform diagram. The measurement signals were stored in TDMS (technical data management streaming) format through the DAQmx configuration record VI. LabVIEW/MATLAB mixed programming was used to process and fit the output signals according to the results of the evolution equation, and the relaxation time value was directly obtained. Figure 6 shows the front panel of the auto-test system.

## 4. Result and Discussion

In order to find a suitable method to measure longitudinal relaxation time, the π pulse method, saturation recovery method, and industry-standard method were compared. The baffle was used to block the pumping light before measurement, so that the atomic system in the cell was restored to a disordered state. Then, the baffle was removed for light pumping. After some time, the π/2 pulse was applied, and the initial amplitude of the precession signal after the π/2 pulse was recorded. The longitudinal spin growth curve was obtained by changing the length of the time of the optical pumping, and then $T_{1}$ was obtained by fitting. This method is called the saturation recovery method. The industry-standard method mentioned in this paper was the reverse recovery method. The comparison results are shown in Figure 7. The $T_{1}$ values at five temperature points were measured using the three methods, and the results showed that the variance range of the π pulse method was 0.4355–0.8511, the variance range of the saturation recovery method was 0.8716–2.085, and the variance range of the industry-standard method was 0.703–1.235. The error of the π pulse method was smaller than that of the other two methods, because the other two methods were susceptible to light instability, resulting in a deviation in the measurement of signal amplitude, and a smaller deviation resulted in a larger error of fitting results.

The automated test system used the pulse method to measure the longitudinal relaxation time and used the FID to measure the transversal relaxation time. The expected temperature was set as 110, 115, 120, 125, and 130. Temperature controls and parameter settings of the main magnetic field are shown in Figure 6. The sweep frequency range of the resonance frequency was set as 100–120 Hz, 30–40 Hz, whereas the step length was set as 1 Hz, the second step length was set as 0.1 Hz, and the sweep frequency range was set as 0.8 Hz. The small excitation amplitude was set as 0.1 V, the pulse amplitude was set as 2.1 V the reading sampling rate was set as 10,000, the sampling number of transverse relaxation time was set as 160,000, and the sampling number of longitudinal relaxation time was set as 400,000. Figure 8 shows the transverse relaxation time of X129e at different temperature points, and Figure 9 shows the longitudinal relaxation time of X131e. It can be seen from the figures that the test data results were basically consistent with the simulation results of the GUI, which proves that there is no problem with our test method and that our test system is correct.

According to the Reference [18], we estimated that the unusual thermal dependence in $T_{1}$ for X 129e could be due to relaxation in the inhomogeneous field. Our team will further research the specific causes of relaxation dependence on temperature in the future. The relaxation mechanism is assumed to be quadrupole coupling between the X 131e nucleus and the electrical field gradients which appear at the nucleus during the adsorption of the atom on the wall. The nuclear quadrupole interaction is assumed to be a perturbation of the magnetic resonance [19]. Figure 10 show the comparison results of the manual test and automatic test system. The automatic test relaxation curve was basically completely consistent with the manual test relaxation curve. The error of $T_{1}$ was within 1.34% and the error of $T_{2}$ was within 0.13%, indicating that the test results of the automatic test system of relaxation time were basically the same as those of the manual test at different temperature points, which proves that the test results are accurate and reliable for the automatic test system of relaxation time of vapor cells.

This test system took 5 h to measure five temperature points, and the manual test took 9.25 h. If we used the industry-standard method, the time of manual testing was 14.25 h, which indicates that the π pulse method is more efficient than the traditional inversion recovery method for measuring $T_{1}$. As it takes a period of time for the Xe tomic group to return to a stable state, we needed to wait at least $5\;T_{1}$ between each delay run. Each data point may take about 10 min to collect, and the inversion recovery method requires multiple data points to improve the fitting accuracy, requiring at least 1 h to measure $T_{1}$. The π pulse method can directly obtain the longitudinal relaxation time using a single measurement, which greatly reduces the time needed for the test, thereby improving the test efficiency. The measuring system developed in this paper is 85% more efficient than the manual test and 185% more efficient than the industry-standard method.

## 5. Conclusions

Based on the principle of relaxation time measurement, this paper built an optical test platform and used a GUI to carry out a visual simulation. The test performance of the π pulse method, saturation recovery method, and industry-standard method in measuring $T_{1}$ was compared. The π pulse method and FID performed best and allowed automating the test process. A user only needs to press the button, and the test results of transverse relaxation time and longitudinal relaxation time can be automatically obtained after a while. The system is easy to operate, which improves the test efficiency, and it works reliably and stably. The test results obtained from the experiment showed that the measurement results of the automated test system were accurate and reliable. The automated test system is currently functional with a stable operation process and a beautiful software interface, which meets the requirements of practical application.

## Figures and Tables

**Figure 1 sensors-19-04863-f001:**
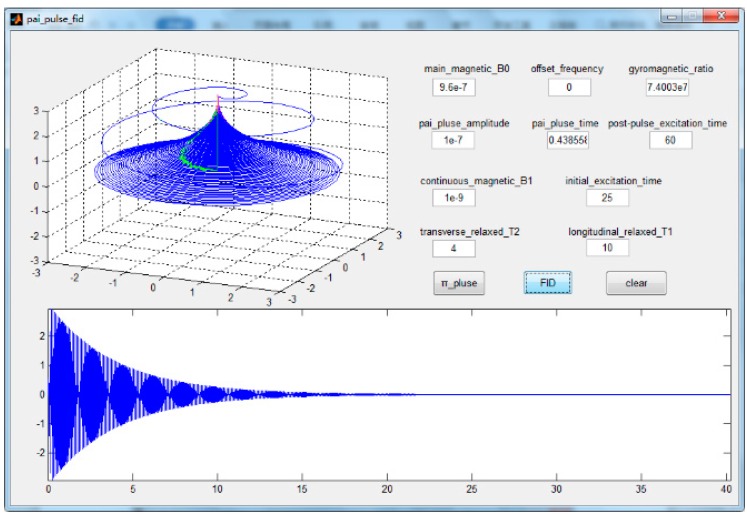
The magnetic moment motion process in the free induction decay (FID) measurement of $T_{2}$. After the π/2 pulse, the magnetic moment has free induction decay, because of atomic collision and other factors.

**Figure 2 sensors-19-04863-f002:**
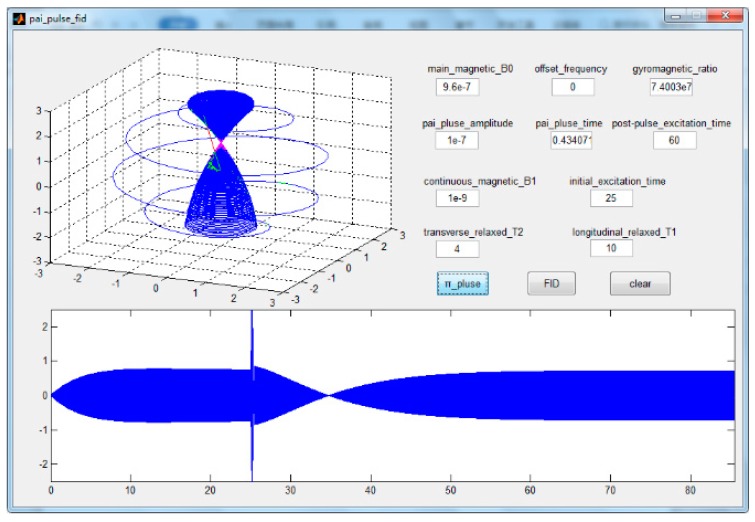
The magnetic moment motion process in the π pulse measurement of $T_{1}$. The first 25 s represent a small excitation process, and the following 60 s represent the motion process of the magnetic moment after the π pulse.

**Figure 3 sensors-19-04863-f003:**
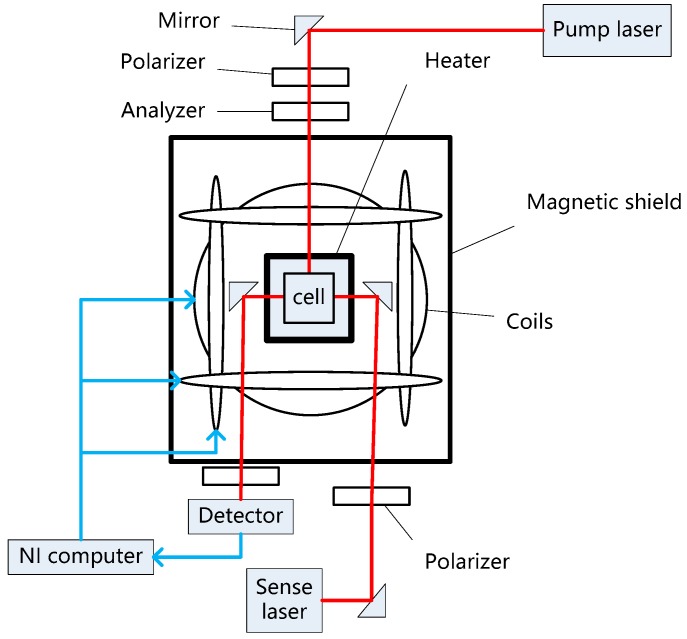
Schematic of the experimental set-up. The red line is the light path and the blue line is the current. The temperature of the vapor cell is controlled by the NI computer.

**Figure 4 sensors-19-04863-f004:**
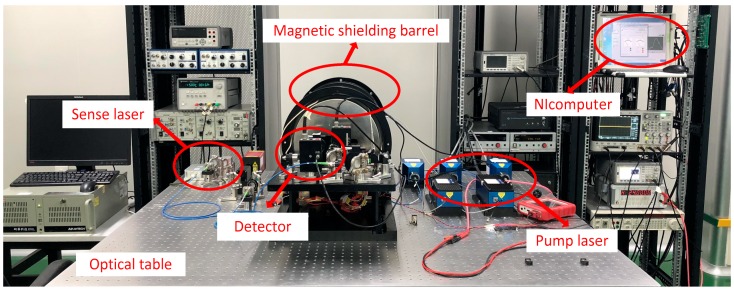
Experimental set-up.

**Figure 5 sensors-19-04863-f005:**
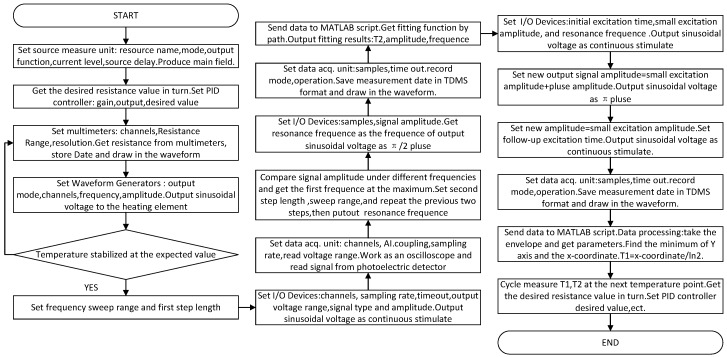
Flow chart of the program.

**Figure 6 sensors-19-04863-f006:**
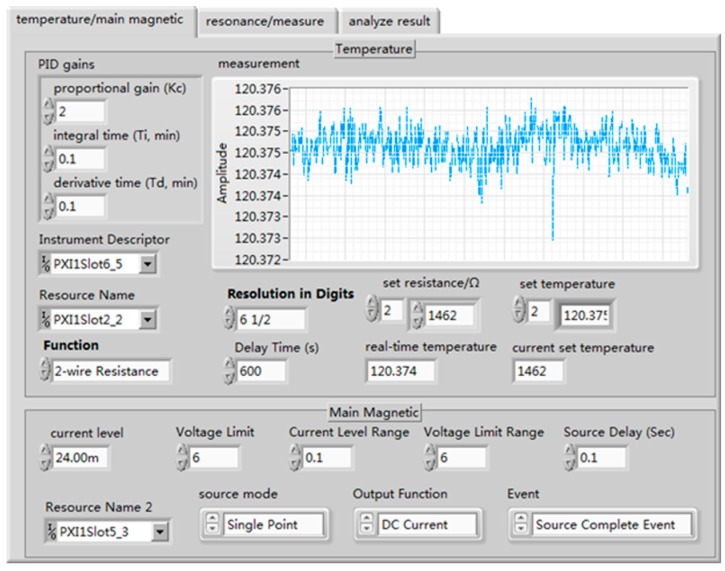
Front panel.

**Figure 7 sensors-19-04863-f007:**
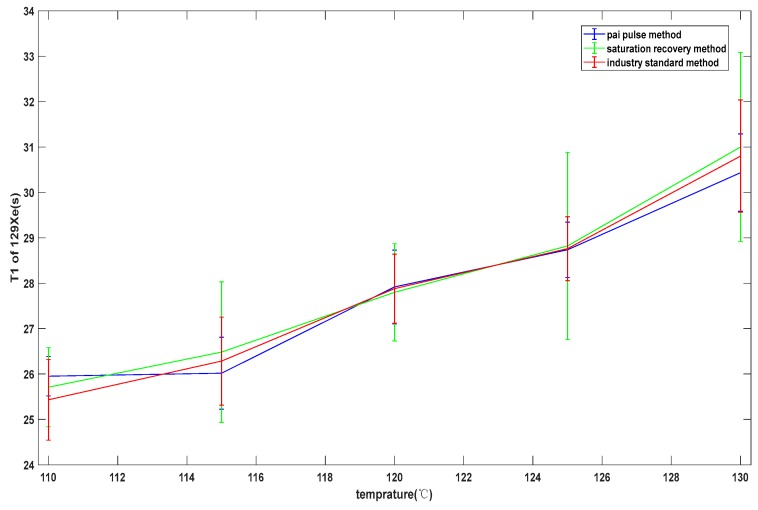
Comparison of test results of the three methods. The blue line shows the test results of the π pulse method. The green line shows the test results of the saturation recovery method. The red line shows the test results of the industry-standard method.

**Figure 8 sensors-19-04863-f008:**
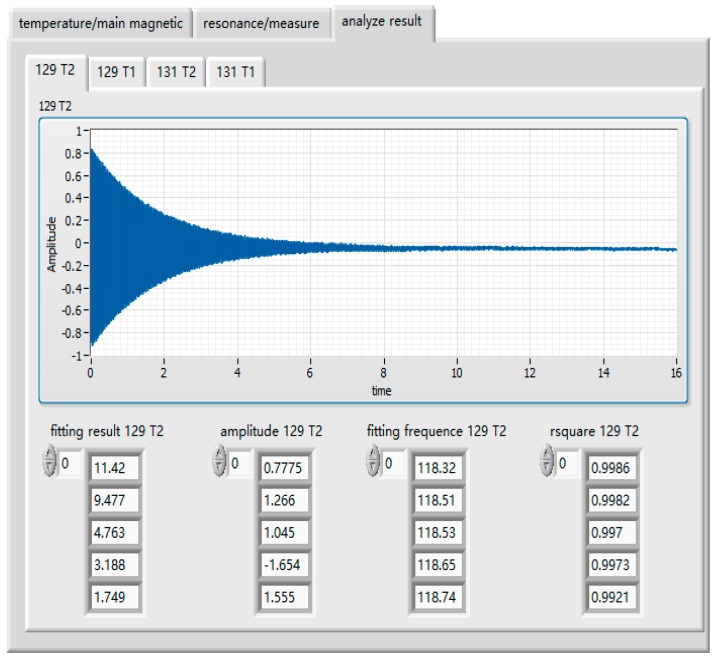
Transverse relaxation time test results of X129e. The determination coefficients are all close to 1, indicating excellent fitting results.

**Figure 9 sensors-19-04863-f009:**
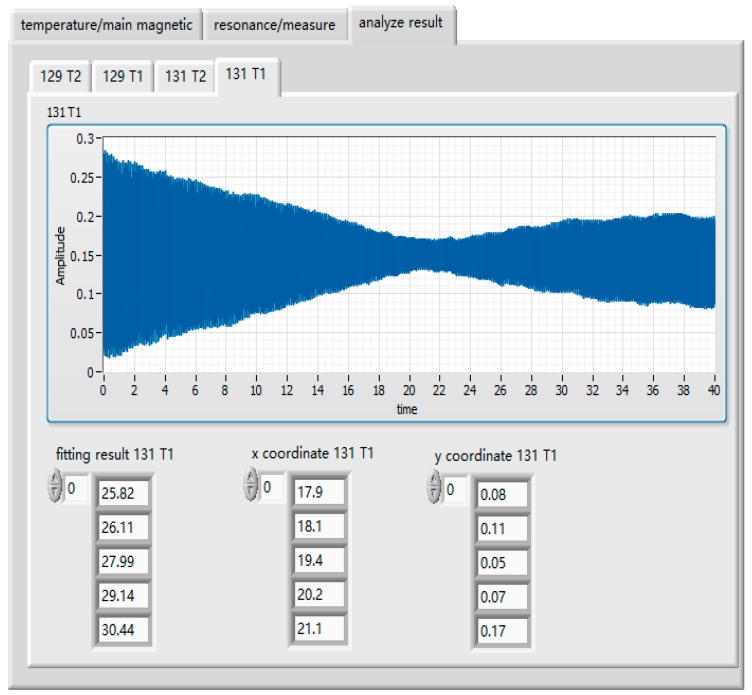
Longitudinal relaxation time test results of X131e. The *x*-coordinate mean half-life $T_{1/2}$ allowed us to get longitudinal relaxation times according to the Equation (8).

**Figure 10 sensors-19-04863-f010:**
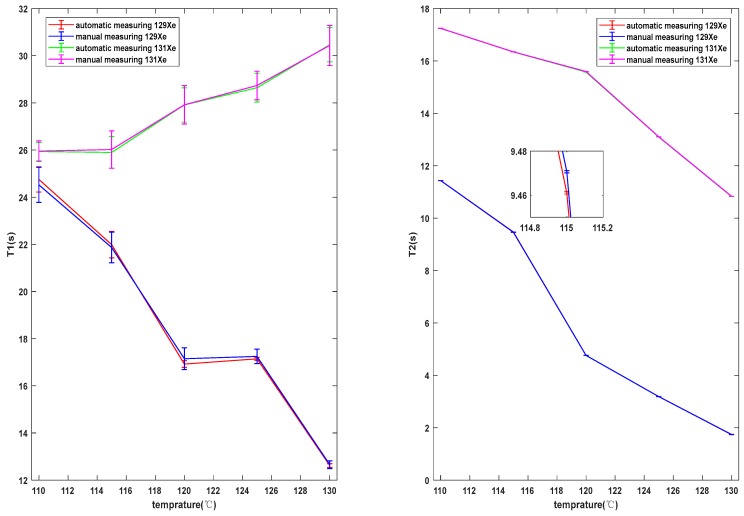
Comparison of test results. The red line shows the test results of X 129e measured by the automatic testing system. The blue line shows the test results of X 129e measured by the manual test. The green line shows the test results of X 131e measured by the automatic testing system. The magenta line shows the test results of X 131e measured by the manual test. The test results for longitudinal relaxation times are on the left, and the test results for transversal are on the right.

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
