# Peer review of "A Fast and Efficient Measurement System for Nuclear Spin Relaxation Times in Atomic Vapors"

_sensors, 2019, doi:10.3390/s19224863_

Round 1

Reviewer 1 Report

Authors describe a nice automated experimental setup with many screenshots. But only this is insufficient to publish a scientific journal paper:  I did not learn anything from reading this manuscript.  

Because the paper is mostly focused at the setup, the title does not correspond to the paper content.

The title shall contain word “nuclear” in “nuclear spin relaxation times.”  

As far as it concerns for the theory part or for the experimental results obtained on the setup, I have not seen any novelty.  A discussion on benefits of π-pulse method vs other methods for T1 measurements deserves dipper elaboration.

A discussion on different temperature dependence of T1 and T2 in two Xe isotopes shall be added.

Test cell shall be specified: size, shape, wall material, coatings, buffer gas, Xe and buffer pressures.

Probe laser shall be specified: targeted Cs transition and eventual detuning from the resonance.

Line 32: What is “precision index” of a gyro? I have never seen a definition.

Description of the probe laser detection in the text does not match the figure: there are two detectors in the text while only one is shown in the figure.

It is not clear how the heating plate driven by current source is located to not disturb the magnetic field.

There is a lack of information regarding the coils (turns or field vs current).

There are many acronyms used without definition,  e.g. TDMS in line 202.

Line 195 . What is the "photoelectric detector"?  Google reports that this is a smoke alarm detector, which has no meaning in the context of this manuscript.  

Line 211: Frequency cannot be measured in volts.

Line 223:  There is a typo in reference to Figure 9 (and not 8).

Lines 234 to 243 need clarifications: no reference is given about “industry standard”.  A comparison with this “industry standard” is also required.

Author Response

Dear editor:

Thank you for your letter and the reviews’ comments concerning our manuscript entitled “A Fast and Efficient Measurement System on Nuclear Spin Relaxation Times in Atomic Vapors”. Those comments are all valuable and very helpful for revising and improving our paper, as well as the important guiding significance to our researches. We have revised our manuscript after reading the comments provided by two reviewers. We appreciate for editors and reviewers’ warm work and hope that the correction will meet with approval. Once again, thank you very much for your comments and suggestion.

Answers to reviewers:

Reviewer #1:

As for the “A discussion on benefits of π-pulse method vs other methods for T1 measurements deserves dipper elaboration.” We added a comparison of π pulse method, saturation recovery method and industry-standard method. The comparison results are shown in figure 7. The error of πpulse method is smaller than that of the other two methods, because the other two methods are susceptible to light instability, resulting in a deviation in the measurement of signal amplitude, and a smaller deviation will result in a larger error of fitting results.

Figure 7. Comparison of test results of three methods. The blue line is the test results of the π pulse method. The green line is the test results of the saturation recovery method. The red line is the test results of the industry-standard method.

As for the “A discussion on different temperature dependence of T1 and T2 in two Xe isotopes shall be added.” The of  is mainly affected by temperature. With increased temperatures come improved alkali pressures and higher particle velocities, increasing collision rate and thus decrease . As the collision rate increases, the particle loses its coherent, resulting in smaller  . The  of  is mainly affected by wall collision rate. The lower the temperature, the higher the time of adsorption when colliding with the wall of the cell, and the  decreases with the decrease of temperature. As for the specific information about the hardware, the experiment adopted 4 mm square borosilicate glass cell with no coating, buffer gas and 1 atmospheric buffer pressure. Pump laser and sense laser adopt American New Focus commercial laser. The working frequency of the pump laser is the resonance frequency of the  atom  line (894nm) and  line (852nm). Sense laser frequency detuning from the optical resonance frequency,and detuning is 0.2nm. This text has only one detector, which receives two optical input signals, and its internal processing is differential. The coil constant of three-dimensional magnetic coil is 24nT/mA. Domestically independently developed non-magnetic heating unit reduces the disturbing magnetic field caused by electric heating by means of precise spatial symmetry arrangement of cables and other means. As for the “There are many acronyms used without definition, e.g. TDMS in line 217.”, we have added the description of the acronyms, and some of the functions that called the program, which have no specific definition, but the code names of the functions. Also, we had modified some of the mistakes in the expression. This statement in line 32 may be a bit of a problem, and what I want to say is the precision of gyroscope directly affects the precision of navigation. We corrected the typo in reference to Figure 10. As for the “photoelectric detector”, photoelectric detector is used to detect the polarization of light signals and may be called detectors or balance detectors. As for the “frequency cannot be measured in volt”, This is a typo, not frequency but excitation amplitude in line247. As for the comparison with this “industry standard”, we shown the measurement results of in figure 7, and the industry standard method of  is FID method.

Reviewer 2 Report

Dear Authors,

your paper is written well.

I have two remarks concerning definitions. First is "horizontal component Mxy" placed on line 97. Perhaps, "transversal component" is better. Second is "the hyperpolarize of spin" placed on line 165. Perhaps "hyperpolarization of spin" is more suitable.

As for I understand, your main acievement is automatization of the relaxation test measurement based on application of data aquisition card together with LabView/Matlab programming. But, I would say, it is a standard way for modern experiment design.

You do not describe both lasers at all. It would be interesting to know the trademark  of the laser, details of their stabilization system, sense laser frequency detuning from the optical resonance frequency. It is worth to give information about main source of signal noise. How stable is direct magnetic field? How precise is realization of condition (6) and (7)? What is the coefficient of attenuation laboratory magnetic field by magnetic shield barrel?

You mention on line 98 that the main influencing factor is inhomogenity of direct magnetic field. So, what is the inhomogenity for you setup?

You compare manual and automatic regime of testing and give an errors for T1 and T2. I suppose the errors characterize deviation of measured value of relaxation time from one regime to another regime. But I do not see error bars for each regime. What is accuracy of the measurement into each regime?

If you recommend your approach for relaxation time measurement, would you so kind to characterize suitable size of the tested cells, the shortest and the longest relaxation time which could be measured with your setup?

In general, I would say that I do not find new very interesting scientific information in your paper. But, the paper could be a very good source of technical information to build similar  setup for relaxation time measurements.

Author Response

Dear editor:

Thank you for your letter and the reviews’ comments concerning our manuscript entitled “A Fast and Efficient Measurement System on Nuclear Spin Relaxation Times in Atomic Vapors”. Those comments are all valuable and very helpful for revising and improving our paper, as well as the important guiding significance to our researches. We have revised our manuscript after reading the comments provided by two reviewers. We appreciate for editors and reviewers’ warm work and hope that the correction will meet with approval. Once again, thank you very much for your comments and suggestion.

Reviewer #2:

As for the two remarks concerning definitions, we modified the definitions. As for the specific information about the hardware, pump laser and sense laser adopt American New Focus commercial laser. Sense laser frequency detuning from the optical resonance frequency,and detuning is 0.2nm. The working frequency of the laser is the resonance frequency of the atom  line (894nm) and  line (852nm). The main source of signal noise is the instability of light. The error of the main magnetic field produced by three-dimensional magnetic coil is within 700pT. The coefficient of attenuation by magnetic shield barrel is 104. As for the realization precise of condition (6) and (7), the condition (6) and (7) get a value 0.11 seconds less than the value measured in the experiment. Keep the current constant so as to keep the magnetic field constant and change the time of excitation pulse. The initial amplitude of FID signal is measured to obtain a series of data points. The data points show sinusoidal distribution with the pulse time length, and the time point at the position of is obtained through fitting, namely the time length of   As for the inhomogeneity of our setup, the magnetic field inhomogeneity is caused by the residual magnetic field gradient in the atomic cell. As for the suitable size of the tested cells, as long as the cell can be measured by FID and π pulse method can be measured with my device, all that needs to be changed is some of the measurement parameters set during the test. We added error bars for each regime which shown as follow:

Figure 10. Comparison of test results. The red line is the test results of  measured by the automatic testing system. The blue line is the test results of  measured by the manual test. The green line is the test results of  measured by the automatic testing system. The magenta line is the test results of  measured by the manual test. The test results for longitudinal relaxation times on the left and the test results for transversal on the right.

Round 2

Reviewer 1 Report

In this revised version, English remains at the limit of being barely understandable.

On the technical side, authors attempted to respond most of my comments (and also most fo the comments of another Referee) and provide missing details. However there are still missing information bits and imperfections that require mandatory minor revision:  

It is not clear to me how the cell can be of “square” shape. Do authors actually mean a cubic shape? Authors still do not provide Xe 131 and Xe 129 pressures in the cell. These pressures shall be specified.   As requested by other Referee, Authors shall provide  a value of the magnetic field gradient in the cell (see the reason in the next point below)     I am very surprised to read the explanation that T1 in Xe 129 is defined by collisions in the cell while T1 in Xe 131 is defined by wall collisions. This is stated without providing citations or supporting estimates and appears to me not justified:

-There is no difference in diffusion coefficients of the two Xe isotopes (in N2 buffer). Therefore it is impossible that collisions in the cell dominate for one isotope, while wall collisions are more important for another isotope;

-The rates of wall collisions and collisions in the cell increase with temperature. None of these mechanisms could explain increase of T1 with with the temperature.

Authors shall support their statement with estimates [e.g. Volk et al, PRL 44, 136 (1980)]. In addition authors should discuss the role of quadrupole effects in Xe 131 [e.g. Donley et al,  PRA 79, 013420 (2009)] and estimate if the unusual thermal depends in T1 for Xe 129 could be due to relaxation in inhomogeneous field [e.g. Cates et al, PRA 37, 2877 (1988).]

Author Response

Thank you for your letter and the  comments of reviewer concerning our manuscript entitled “A Fast and Efficient Measurement System on Nuclear Spin Relaxation Times in Atomic Vapors”. Those comments are all valuable and very helpful for revising and improving our paper, as well as the important guiding significance to our researches. We have revised our manuscript after reading the comments provided by the reviewer. We appreciate for editors and warm work of reviewer and hope that the correction will meet with approval. Once again, thank you very much for your comments and suggestion.

Round 3

Reviewer 1 Report

Authors responded all technical comments and remarks. From this point of view, the paper can be published. However English can be largely improved.